# The Feasibility and Acceptability of Neurologic Music Therapy in Subacute Neurorehabilitation and Effects on Patient Mood

**DOI:** 10.3390/brainsci12040497

**Published:** 2022-04-13

**Authors:** Naomi Thompson, Jodie Bloska, Alison Abington, Amber Masterson, David Whitten, Alexander Street

**Affiliations:** 1Cambridge Institute for Music Therapy Research, Anglia Ruskin University, Cambridge CB1 1PT, UK; jodie.bloska@aru.ac.uk (J.B.); alex.street@aru.ac.uk (A.S.); 2The Marbrook Centre, St Neots PE19 8EP, UK; alisonabington@marbrook.co.uk (A.A.); ambermasterson@marbrook.co.uk (A.M.); davidwhitten@marbrook.co.uk (D.W.)

**Keywords:** neurologic music therapy, subacute neurorehabilitation, feasibility, acceptability, mood

## Abstract

Music interventions support functional outcomes, improve mood, and reduce symptoms of depression in neurorehabilitation. Neurologic music therapy (NMT) has been reported as feasible and helpful in stroke rehabilitation but is not commonly part of multidisciplinary services in acute or subacute settings. This study assessed the feasibility and acceptability of delivering NMT one-day-per-week in a subacute neurorehabilitation centre over 15 months. Data were collected on the number of referrals, who referred, sessions offered, attended, and declined, and reasons why. Staff, patients, and their relatives completed questionnaires rating the interventions. Patients completed the Visual Analog Mood Scales (VAMS) pre and post a single session. Forty-nine patients received 318 NMT sessions (83% of sessions offered). NMT was rated as helpful or very helpful as part of the multidisciplinary team (*n* = 36). The highest ratings were for concentration, arm and hand rehabilitation, and motivation and mood. VAMS scores (*n* = 24) showed a reduction in ‘confused’ (−8.6, *p* = 0.035, effect size 0.49) and an increase in ‘happy’ (6.5, *p* = 0.021, effect size = 0.12) post NMT. The data suggest that a one-day-per-week NMT post in subacute neurorehabilitation was feasible, acceptable, and helpful, supporting patient engagement in rehabilitation exercises, mood, and motivation.

## 1. Introduction

The value and potential for music interventions, including neurologic music therapy (NMT), within multidisciplinary neurorehabilitation have been recognised and discussed [1,2,3,4,5,6,7]. However, few music therapists are currently working in neurorehabilitation internationally [8,9]. There is a lack of implementation research looking at the feasibility and acceptability of music interventions within existing multidisciplinary services, which traditionally include physiotherapists, occupational therapists, and speech and language therapists. One study on an acute stroke ward found a two-day-per-week NMT service to be feasible and helpful for mood and motivation while simultaneously supporting functional recovery [10].

Music interventions with brain-injured populations may help to improve gait parameters, timing of arm movements, aphasia, and quality of life [11]. Studies have shown music interventions in subacute neurorehabilitation to support gait training [12,13], upper extremity [14,15,16], speech [17], and cognitive recovery [18].

Previous research on NMT indicates a positive effect on mood, quality of life, and well-being, and a reduction in symptoms of depression and anxiety, while addressing functional goals [19,20,21,22]. Music interventions may be less fatiguing than traditional therapies [16,23,24], possibly due to neurochemical changes induced by music [25], enhanced levels of neuroplasticity [15,26], and the activation of neural mechanisms involved in emotion regulation, motivation, reward, and arousal [6]. This could support adherence to rehabilitation programmes and so help patients meet dosage requirements, which is crucial for effective recovery [27,28,29].

This study looked at the feasibility and acceptability of a one-day-per-week NMT post in a subacute neurorehabilitation centre over a 15-month period.

## 2. Materials and Methods

### 2.1. Setting

The NMT worked within the multidisciplinary team (MDT) for 7.5 h per week. The centre also hosted a music therapy master of arts (MA) student on placement for 10 h per week across eight months of the service evaluation. Two MA students were hosted at the centre, one of whom had completed NMT training. Both were supported to use NMT techniques through weekly supervision with NMT trained music therapists.

During the setup of the post, a presentation outlining the mechanisms of NMT, including video demonstrations and clarifying referral criteria, was delivered to the MDT. A therapy assistant working at the centre was assigned to liaise between the MDT and the music therapist. Members of the MDT regularly attended a sensorimotor group run by the music therapist as well as conjoint working in individual sessions.

### 2.2. Data Collection

Data were collected as part of usual care, including: age, sex, reason for referral to NMT, number of sessions offered and attended, reasons for non-attendance, type of session, interventions used, and who the referral was from. Patients were referred verbally or by e-mail for NMT by members of the MDT. Data on attendance, alongside feedback from the questionnaires, were used to evaluate the feasibility of delivering NMT in a one-day-per-week post. There was no control group as the study aimed to evaluate the feasibility of NMT as part of MDT subacute neurorehabilitation.

All data collection tools were approved by the centre manager and clinical lead, and permission to collect data as part of standard care and for publishing was granted by the rehabilitation centre. The music therapist and MA student explained the reason and purpose of data collection to each patient, including that data would be anonymised, used for a service evaluation, and possibly published. Each patient then provided verbal consent.

The data collection tools comprised three questionnaires (staff, patients, patients’ relatives) and the Visual Analog Mood Scales (VAMS) [30]. These were chosen to align with tools used in a recent study evaluating the feasibility of NMT in an acute stroke ward [10]. A version of the patient questionnaire for those with aphasia was co-designed with the speech and language therapist. The questionnaire was administered to patients after they had attended one or more sessions by other members of staff, not the music therapist, to reduce bias. Members of the MDT and relatives were asked to complete a questionnaire after they had attended a session. The questionnaires asked respondents to rate whether they thought NMT was 1. Not Helpful, 2. Quite Helpful, 3. Helpful, 4. Very Helpful, or 5. Not Applicable towards five therapeutic aims: ‘speech and communication’, ‘walking and mobility’, ‘concentration’, ‘arm and hand movement’, and ‘motivation and mood’. There was space for additional comments with prompting questions.

VAMS [31] data were collected by the music therapist and other staff pre and post a single NMT session. The lead music therapist was qualified to deliver VAMS per the required standard and supervised students and other staff in its use. Eight mood states are included in the VAMS: six negative (afraid, confused, sad, angry, tired, tense) and two positive (energetic, happy). Data were not collected in the first session to allow the patient to adjust to the new therapy input.

### 2.3. Statistical Analysis

*T* scores were generated from the VAMS raw scores and used for statistical analysis. For identifying mood disorders, a *T*-score of <60 for negative emotions or >40 for positive emotions is considered significant [31]. When tracking changes in mood, differences between the pre- and post-*T* scores above 20*T* are considered reliable, and differences over 30*T* are considered clinically significant. SPSS 26 was used to generate means for pre- and post-session *T* scores along with standard deviation and confidence intervals. As the data were not normally distributed, a Wilcoxon signed ranks test was conducted, as well as an effect size calculation (Cohen’s d). A significance level of 0.05 was used with 95% confidence intervals.

The questionnaire generated both quantitative and qualitative data. A thematic analysis was conducted on additional comments with written responses coded using inductive coding and grouped into themes.

### 2.4. Intervention

Many of the interventions used were standardised NMT exercises to achieve motor, cognitive, and speech and communication goals in line with those set by the MDT [32]. These included a weekly sensorimotor group using therapeutical instrumental music performance (TIMP), where patients played instruments in time with music provided by the music therapist to target specific movements. This took place in a private lounge at the centre. Individual sessions also used TIMP, as well as other NMT techniques such as patterned sensory enhancement (PSE), melodic intonation therapy (MIT), rhythmic speech cueing (RSC), vocal intonation therapy (VIT), musical neglect training (MNT), and musical attentional control training (MACT). Additionally, the music therapist and MA student supported the psychological and emotional well-being of patients through techniques such as improvising music, singing meaningful familiar songs, and songwriting. Individual sessions took place in a private lounge or in patient rooms.

## 3. Results

### 3.1. Demographic Data

Over 15 months, 49 patients received NMT. A total of 24 patients (49% of total patients seen) who attended NMT completed the mood scale, while 22 completed the questionnaire (45% of total patients seen, see Figure 1). Reasons for non-completion included limited staff time, patients being unable to complete the questionnaires due to cognitive impairments, patients not consenting to take part, and patients being discharged before it was possible to complete the questionnaires. Nine members of the MDT and five patient relatives completed the questionnaire. The majority of patients who completed the forms attended a sensorimotor group (71%) that focused on patients’ physiotherapy and occupational therapy goals in addition to music therapy goals. A total of 22% of these patients also had individual NMT, while 27% received individual NMT only. The average age of patients attending NMT was 68.7 ± 17.5 years. The mean time between the referral and the first session was 10.2 ± 7.9 days. The mean time between injury onset and the first NMT session was 137.6 ± 108.8 days. A total of 49% of patients (29) had sessions with the music therapist, 10% (6) had sessions with an MA student, and 41% (24) were seen by the therapist and student. Table 1 displays the demographic data and diagnoses.

### 3.2. Questionnaires

All staff and patient relatives, and the majority of patients (86.4%), rated NMT as helpful or very helpful in neurorehabilitation and as part of the MDT (Table 2). For all participants, the areas in which NMT was rated most helpful were ‘concentration’, ‘arm and hand rehabilitation’, and ‘motivation and mood’. ‘Speech and communication’ and ‘walking and mobility’ were the variables most often rated as not applicable (Table 3).

Eighteen patients provided qualitative feedback. Most responses to ‘Did music therapy help with anything else?’ came under the theme of ‘mood’ (10, 55.6%), ‘movement’ (6, 33.3%), or ‘social’ (4, 22.2%), with some responses being categorised in more than one theme. Other themes were ‘cognition’ and ‘motivation’. Two patients suggested a different music selection for the group when asked ‘Is there anything you would like to have been included in the music therapy sessions?’ (see Appendix A, Table A1). Seven staff members provided comments, with the majority coming under the themes ‘social’ (6, 85.7%) and ‘mood’ (3, 42.9%). One staff member suggested that the music group might be too long (see Appendix A, Table A2). Three patient relatives provided qualitative feedback, with two (66.7%) being categorised as ‘mood’ and ‘social’ and one (33.3%) coming under the theme ‘movement’ (see Appendix A, Table A3).

### 3.3. Visual Analog Mood Scale

The results from the VAMS show that, on average, the mood states of those who participated in NMT were in the normal range for both positive and negative moods pre and post a single NMT session (Table 4, Figure 2) [28].

## 4. Discussion

Over the period reported, based on the number of referrals, sessions offered and completed, and questionnaire data, the service was feasible with the additional input from the MA student and the liaison person. The exercises were acceptable and considered helpful within neurorehabilitation, in particular, toward goals relating to concentration, arm and hand movement, motivation, and mood. VAMS scores showed positive changes in seven of eight mood variables after a single session, with a statistically significant increase in ‘happiness’ and a decrease in ‘confusion’.

This study looked at the feasibility and acceptability of delivering NMT one-day-per-week in one neurorehabilitation centre. Results cannot be generalised to other services; however, there are similarities with previous research in acute stroke rehabilitation in regard to patient engagement and perceived benefit of the interventions [10].

As data were collected as a service evaluation, there are several limitations. No comparison data were collected, so it is not possible to say whether a similar questionnaire and VAMS data would have resulted from other MDT sessions. Of a total of 49 patients who received NMT, 24 completed the mood scale, and 22 completed the questionnaire. All members of the MDT who jointly ran NMT sessions completed the staff questionnaires, totalling 9 of 10 members of staff. One member of the MDT, a psychologist, worked on a different day to the NMT, so it was not possible to do conjoint work. There may have been some bias in the data collection, as data were collected by clinicians working at the centre. However, in an attempt to mitigate this, staff other than the NMT collected all patient questionnaire data and some VAMS data. The questionnaire data show participants’ perception of the benefits of functional goals, but no data were collected on functional outcomes using validated measures.

The questionnaire data suggest that NMT was considered most helpful for ‘concentration’, ‘arm and hand rehabilitation’, and ‘motivation and mood’. Although ‘speech and communication’ or ‘walking and mobility’ were reported as not applicable for 15 (34%) patients, NMT was still reported as helpful or very helpful for addressing these goals for 24 (55%) patients. This pattern was seen in a previous study [10], which used the same feedback forms in an acute stroke setting. Although exercises for lower limb and walking phases were included in the sensorimotor group, patients may not have comprehended that these were linked to gait. Thematic analysis of the additional questionnaire comments showed that NMT raised patient mood and provided an opportunity for social interaction as well as supporting sensorimotor rehabilitation, which is supported by previous study findings [16,22].

There was more variety in questionnaire ratings from patients than from their relatives and staff. This could be due to patients being more focused on the applicability to their own goals, while their relatives and clinicians might consider the potential effectiveness across more domains and recognise the importance of motivation for patients to engage sufficiently in rehabilitation. Additionally, clinicians would have a greater understanding of the mechanisms underlying the interventions through their own clinical knowledge, the initial presentation given to the MDT prior to the NMT commencing, and additional informal communications with the music therapist. However, patients may have an initial preconception of music as a recreational activity. Future research could explore the role awareness of the mechanisms underlying NMT has in clinician and patient perceptions of the helpfulness of the interventions.

The VAMS scores support the subjective questionnaire results showing positive changes in mood states pre and post a single session. The significant reduction in levels of confusion could suggest that engaging in music interventions improves cognitive functioning and orientation, as suggested in the literature [16,18,19]. The increase in ‘energetic’, despite the nature of the exercises, supports the suggestion that engaging in music-based exercises is less fatiguing [16,23,24].

## 5. Conclusions

Data suggest that NMT was feasible within the subacute setting at one-day-per-week. It was considered helpful, especially for movement, concentration and motivation, and mood, with the latter supported by changes in mood scores (VAMS). Integrating the music therapist into the MDT through conjoint working and having a named, full-time member of staff to facilitate the service was invaluable to the referral system setup for a one day post. This also helped staff to quickly understand the mechanisms of the interventions. Future research should be conducted to determine whether music-based interventions increase patient engagement in other MDT interventions due to the social and mood benefits and whether this improves clinical outcomes, reduces inpatient length of stay, and increases patient flow.

## Figures and Tables

**Figure 1 brainsci-12-00497-f001:**
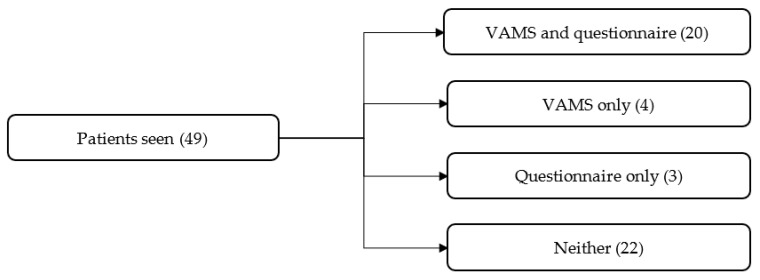
Patient participation in Visual Analog Mood Scale (VAMS) and questionnaire feedback.

**Figure 2 brainsci-12-00497-f002:**
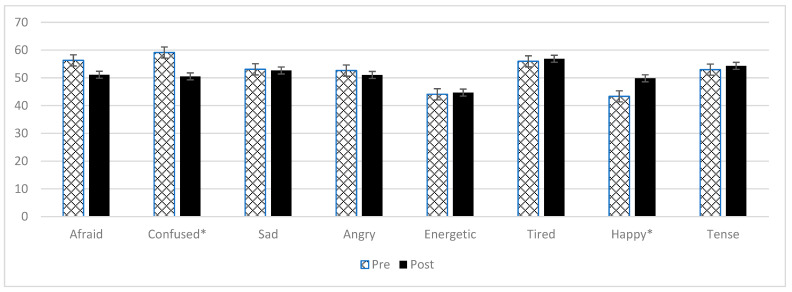
VAMS scores pre and post a single NMT session. Mood states marked with * show a statistically significant change.

**Table 1 brainsci-12-00497-t001:** Patient demographic and diagnostic data, and referral reason and attendance at Neurologic Music Therapy (NMT).

**Sex**	Male	25 (51%)
	Female	24 (49%)
**Diagnosis**	Stroke	32 (65%)
	Haematoma	3 (6%)
	Leukoencephalopathy	2 (4%)
	Spinal cord injury	3 (6%)
	Brain or spinal cord tumour	3 (6%)
	Other *	6 (12%)
**Type of Session**	Group	25 (51%)
	Individual	14 (29%)
	Both	10 (20%)
**No. of Sessions**	Offered	382
	Attended	318 (83%)
	Declined	29 (8%)
	Unable	35 (9%)
**Referred By**	PT	9 (18%)
	OT	7 (14%)
	SLT	5 (10%)
	PT/OT	20 (41%)
	OT/SLT	1 (2%)
	PT/OT/SLT	3 (6%)
	TA/OT	1 (2%)
	Psychologist	0 (0%)
	MT	0 (0%)
	Patient request	1 (2%)
	Not stated	2 (4%)
**Referral for Goals Relating to ****	Cognition	9 (18%)
	Communication	11 (22%)
	Emotional expression	7 (14%)
	Gait	4 (8%)
	Lower limb	28 (57%)
	Upper limb	38 (78%)
	Mood	2 (4%)
**NMT Intervention**	TIMP	35 (71%)
	Other	2 (4%)
	TIMP + other	7 (14%)
	Not stated	5 (10%)

* Diagnoses in ‘other’ category include: pneumococcal meningitis, brain aneurysm, hydrocephalus, skull fracture, and subsequent stroke, duodenal ulcer, and heart attack. ** Most referrals were for more than one reason, and so percentage is shown as % of total no. of patients. PT—physiotherapy; OT—occupational therapy; SLT—speech and language therapy; TA—therapy assistant; MT—music therapist; NMT—neurologic music therapy; TIMP—therapeutic instrumental music performance.

**Table 2 brainsci-12-00497-t002:** Responses to the question ‘What do you think of music therapy?’.

Participants	Questions	Not Helpful (%)	Quite Helpful (%)	Helpful (%)	Very Helpful (%)	Not Applicable (%)
Patients (*n =* 22)	What did you think of the MT session	0	13.6	40.9	45.5	0
Staff (*n =* 9)	What do you think about MT for patients?	0	0	22.2	77.8	0
	What do you think of MT as part of MDT?	0	0	44.4	55.6	0
Relatives (*n* = 5)	What did you think of MT in stroke rehab?	0	0	0	100	0

**Table 3 brainsci-12-00497-t003:** Responses to the questions ‘What did music therapy help with?’.

Did MT Help with:	Speech and Communication	Walking and Mobility	Concentrating	Arm and Hand	Motivation and Mood
Patients (*n* = 22)					
Not helpful (%)	4.5	4.5	0	4.5	0
Quite Helpful (%)	9.1	4.5	9.1	18.2	18.2
Helpful (%)	27.3	36.4	31.8	22.7	27.3
Very Helpful (%)	27.3	18.2	54.5	45.5	40.9
Not applicable (%)	31.8	36.4	4.5	9.1	4.5
No data (%)	0	0	0	0	9.1
Staff (*n* = 9)					
Not helpful (%)	0	0	0	0	0
Quite Helpful (%)	0	11.1	0	0	0
Helpful (%)	66.7	33.3	33.3	22.2	22.2
Very Helpful (%)	11.1	44.4	66.7	77.8	66.5
Not applicable (%)	22.2	11.1	0	0	0
No data (%)	0	0	0	0	11.1
Relatives (*n* = 5)					
Not helpful (%)	0	0	0	0	0
Quite Helpful (%)	0	20	0	0	0
Helpful (%)	40	0	0	20	0
Very Helpful (%)	60	40	100	80	100
Not applicable (%)	0	40	0	0	0

**Table 4 brainsci-12-00497-t004:** Responses to the Visual Analog Mood Scale (*n* = 24).

	Pre-Mean (SD)	Post-Mean (SD)	Difference (95% CI)	*p*-Value	Cohen’s d
Afraid	56.3 (19.9)	51.1 (17.6)	−5.2 (4.4, −14.8)	*p* = 0.081	0.27
Confused	59.1 (20.6)	50.5 (14.0)	−8.6 (−0.2, −17.0)	*p* = 0.035	0.49
Sad	53.1 (15.4)	52.7 (19.6)	−0.4 (8.2, −9.0)	*p* = 0.698	0.02
Angry	52.6 (14.3)	51.0 (17.8)	−1.6 (6.9, −10.0)	*p* = 0.433	0.10
Energetic	44.1 (13.7)	51.0 (17.8)	7 (15.6, −1.7)	*p* = 0.153	0.44
Tired	56.0 (11.5)	56.9 (14.2)	0.9 (7.1, −5.3)	*p* = 0.692	0.07
Happy	43.3 (11.4)	49.8 (10.7)	6.5 (12.3, 0.7)	*p* = 0.021	0.59
Tense	53.0 (16.0)	54.3 (20.2)	1.4 (7.9, −5.1)	*p* = 1.00	0.12

## Data Availability

The data presented in this study are available upon reasonable request from the corresponding author.

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
