# Peer review of "The Feasibility and Acceptability of Neurologic Music Therapy in Subacute Neurorehabilitation and Effects on Patient Mood"

_brainsci, 2022, doi:10.3390/brainsci12040497_

Round 1

Reviewer 1 Report

The manuscript entitled “The Feasibility and Acceptability of Neurologic Music Therapy in Subacute Neurorehabilitation and Effects on Patient Mood” is a fair attempt in the field of therapy against neurological problems. However, I would like to suggest that the study has minor defects.

  1. A lot of grammatical and typological errors appear in the manuscript.
  1. No need to write numbers after each keyword.
  2. In the materials and methods section, what is an MA student?
  3. In the materials and methods section if the authors give the diagrammatic representation of the procedure adopted so it will be more clear for the reader.
  4. The authors have reported the result only as tables, I recommend providing some results as graphs.
  1.  I recommend the authors should include a graphical abstract to make the understanding easy and clearer.
  2. The discussion has to be thoroughly checked and has to be amended so as to make it clearer and more specific.

 On the basis of the above comments, I recommend the minor changes before acceptance.

Author Response

Thank you for your careful review and helpful comments. Please see response to each point below and corresponding changes in the attached revised manuscript.

  1. A lot of grammatical and typological errors appear in the manuscript.

The first, second and last author carefully read the manuscript, editing any errors. These are shown in track changes.

  1. No need to write numbers after each keyword.

Numbers removed after each keyword. (p1)

  1. In the materials and methods section, what is an MA student?

This term was clarified the first time of writing with the abbreviation used thereafter (p2)

  1. In the materials and methods section if the authors give the diagrammatic representation of the procedure adopted so it will be more clear for the reader.

A flow chart for patient participation in mood and evaluation tools has been added for added clarity. (Figure 1, p5)

  1. The authors have reported the result only as tables, I recommend providing some results as graphs.

A figure showing a visual image of the data in Table 4 was added (Figure 2, p9)

  1.  I recommend the authors should include a graphical abstract to make the understanding easy and clearer.

A graphical abstract was created.

  1. The discussion has to be thoroughly checked and has to be amended so as to make it clearer and more specific.

The discussion was checked thoroughly, with clarifications shown in tracked changes (p9-10)

Reviewer 2 Report

The study investigated the feasibility and acceptability of NMT in subacute neurorehabilitation unit, and its effects on patient’s mood. The result showed positive results in improving certain domains of mood.

Here are some points need to be clarified.

Abstract

Line 19-21 “The data suggest that a one-day NMT post in subacute neurorehabilitation was feasible, acceptable and helpful, supporting patient engagement in rehabilitation exercises, mood and motivation.”

I suggest use other term for “one-day NMT post”, e.g., one-day NMT session

Introduction

The introduction provides adequate background to address the effect of music therapy.

Line 42-44

“neurochemical changes induced by music, enhanced levels of neuroplasticity, and the activation of neural mechanisms involved in emotion regulation, motivation, reward and arousal.”  References are needed to point out each mechanism.

Materials and Methods

I suggest use subheading in this section for better understanding.

For example, Participant, Intervention, Data collection, Statistical analysis, etc.

What is the duration since onset of diseases of these patients? The title define the intervention was applied to the subacute neurorehabilitation, but the related information is lacking.

Line 50 and Line 51

Please define the abbreviation of MDT and MA in the first time.

Line 60

“referral reason”, are you meaning “referral diagnosis”?

I suppose the reason may refer to “inattention”, “motor deficit”, “poor progression” ..etc. Since these patients who receive music therapy were referred by the clinicians, I would like to know what condition that make them refer the patient to music therapy rather than the definite diagnosis.

Line 67-69

Was the intervention approved by the local Institution of Review Board?  Please provide the IRB number if applicable

Line 84

Were all the music therapist and other staff being certified or trained for evaluating VAMS for the patient?

Line 96-97

“Categorical data were transformed into percentages in Excel”. The sentence is redundant, please consider removing it.

Line 100-111

This paragraph should being placed in the subheading of “Intervention”, so that the readers can easily understanding the interventions.

Results

Line 120-122

The standard deviation can be revised to 68.7 ± 17.5 years and 10.2 ±7.9 days.

Table 1

The diagnosis should be more simplified. A (n= xx) should be place after the diagnosis.

For example, there are right MCA infarct, as well as right MCA stroke? Are they different?

And all the term should be unified, like haemorrhagic stroke and haemorrhage are same meaning.

Basal ganglia haemorrhage stroke, right/left (n=1/1)

An interesting case is T8 meningioma. Though spinal cord lesion belongs to neurorehabilitation, is the theory of music therapy suggest the mechanism for brain injury can be apply to the spinal cord lesion?

Line 130-131

This information was repeated as Line 114-116. Please simplified or remove either here or before.

Table 4

The statistical method you used here is paired t-test. However, is the data being normal distribution? Why not use Wilcoxin test?

Discussion

Line 156-164

The paragraph should be concentrated. There is no need to place (Table 2) , (Table 4) to illustrate the main result.

Line 168-174

This paragraph may be better if put in the result.

Line 200-202

“Additionally, clinicians would have a greater understanding of the mechanisms underlying the interventions, while patients may have an initial preconception of music as a recreational activity.”

I have concern for this interpretation. As you raise in the study, some of the outcome and phenomenon was observed by the rating scale and questionnaire, however, the neurological mechanisms were not discussed in the manuscript. Was the change can be explained by the neurotransmitter, neuroplasticity or other theory? I understand that this is not the aim of the study, however, this is the weak point why clinicians may consider music therapy as a recreational activity. You may address the issue for future research direction.  

Author Response

Thank you for your thoughtful review and comments on the manuscript. We have addressed each point and feel that this has improved the manuscript. Please see response to each point below:

Abstract

Line 19-21 “The data suggest that a one-day NMT post in subacute neurorehabilitation was feasible, acceptable and helpful, supporting patient engagement in rehabilitation exercises, mood and motivation.”

I suggest use other term for “one-day NMT post”, e.g., one-day NMT session

Line 19 - 22: I have kept the term 'post' as we were looking at the feasibility of integrating a neurologic music therapist into the multidisciplinary team as a whole. However, I have clarified that this was delivered one-day 'a week'.

Introduction

The introduction provides adequate background to address the effect of music therapy.

Line 42-44

“neurochemical changes induced by music, enhanced levels of neuroplasticity, and the activation of neural mechanisms involved in emotion regulation, motivation, reward and arousal.”  References are needed to point out each mechanism.

additional references included (Line 43-45)

Materials and Methods

I suggest use subheading in this section for better understanding.

For example, Participant, Intervention, Data collection, Statistical analysis, etc.

Subheadings were included throughout the materials and methods section (p2-3)

What is the duration since onset of diseases of these patients? The title define the intervention was applied to the subacute neurorehabilitation, but the related information is lacking.

Duration from onset of disease to first music therapy session has been added to the results section (line 134-135)

Line 50 and Line 51

Please define the abbreviation of MDT and MA in the first time.

Terms defined when first appearing (line 52-53)

Line 60

“referral reason”, are you meaning “referral diagnosis”?

I suppose the reason may refer to “inattention”, “motor deficit”, “poor progression” ..etc. Since these patients who receive music therapy were referred by the clinicians, I would like to know what condition that make them refer the patient to music therapy rather than the definite diagnosis.

Thank you for pointing this out. We meant the reason for the referral to NMT - this data has been added to the results in Table 1 (p5-6)

Line 67-69

Was the intervention approved by the local Institution of Review Board?  Please provide the IRB number if applicable

As the data was collected as part of a service evaluation we did not need to go through the local Institution of Review Board. All patients and staff who participated were informed of the purpose and possible use of the data. Access to NMT was not impacted by whether the patient completed the questionnaires.

Line 84

Were all the music therapist and other staff being certified or trained for evaluating VAMS for the patient?

the publisher in the states says the qualification level is: A degree from an accredited 4-year college or university in psychology, counseling, speech-language pathology, or a closely related field plus satisfactory completion of coursework in test interpretation, psychometrics and measurement theory, educational statistics, or a closely related area; or license or certification from an agency that requires appropriate training and experience in the ethical and competent use of psychological tests.

In the UK: Psychologist- Products with a qualification code A are intended for qualified psychologists who have normally completed additional training in their particular specialism.

The lead music therapist met the above requirements and supervised all use of VAMS. This has been included (line 90-91)

Line 96-97

“Categorical data were transformed into percentages in Excel”. The sentence is redundant, please consider removing it.

Sentence has been removed (line 104-105)

Line 100-111

This paragraph should being placed in the subheading of “Intervention”, so that the readers can easily understanding the interventions.

The subheading 'intervention' has been added (line 108)

Results

Line 120-122

The standard deviation can be revised to 68.7 ± 17.5 years and 10.2 ±7.9 days.

Reporting of standard deviation revised (line 132-134)

Table 1

The diagnosis should be more simplified. A (n= xx) should be place after the diagnosis.

For example, there are right MCA infarct, as well as right MCA stroke? Are they different?

And all the term should be unified, like haemorrhagic stroke and haemorrhage are same meaning.

Basal ganglia haemorrhage stroke, right/left (n=1/1)

An interesting case is T8 meningioma. Though spinal cord lesion belongs to neurorehabilitation, is the theory of music therapy suggest the mechanism for brain injury can be apply to the spinal cord lesion?

Diagnosis simplified and conditions grouped to make it simpler for the reader (see revised Table 1 p5-6)

Line 130-131

This information was repeated as Line 114-116. Please simplified or remove either here or before.

Thank you for pointing this out - the repeated information has been removed. (line 150-151)

Table 4

The statistical method you used here is paired t-test. However, is the data being normal distribution? Why not use Wilcoxin test?

Thank you for this comment. We checked the data again and it is not normally distributed so have conducted a Wilcoxon test instead. (line 101-103; table 4, p9)

Discussion

Line 156-164

The paragraph should be concentrated. There is no need to place (Table 2) , (Table 4) to illustrate the main result.

I have edited the paragraph and removed references to tables in the results (line 179-189)

Line 168-174

This paragraph may be better if put in the result.

Reasons for not completing questionnaires were moved to the results section (line 125-128)

Line 200-202

“Additionally, clinicians would have a greater understanding of the mechanisms underlying the interventions, while patients may have an initial preconception of music as a recreational activity.”

I have concern for this interpretation. As you raise in the study, some of the outcome and phenomenon was observed by the rating scale and questionnaire, however, the neurological mechanisms were not discussed in the manuscript. Was the change can be explained by the neurotransmitter, neuroplasticity or other theory? I understand that this is not the aim of the study, however, this is the weak point why clinicians may consider music therapy as a recreational activity. You may address the issue for future research direction.  

Neuroplasticity is the main theory supporting the effectiveness of music therapy in this field. This was explained to the clinicians in the initial presentation to staff, and reiterated in informal communications by the NMT. The discussion here has been clarified and expanded (line 227-233)

Round 2

Reviewer 2 Report

Thank you for the revision. There is only one point I suggest to modify. 

In Table 1, the referral reason includes "cognition", "communication"... etc.

However, the term could be more precise to point out these are the impairment that patients have. For example "Impairment for Referral". 

Author Response

Thank you again for your careful reading of the manuscript. Changes made are outlined below

In Table 1, the referral reason includes "cognition", "communication"... etc.

However, the term could be more precise to point out these are the impairment that patients have. For example "Impairment for Referral". 

The label in Table 1 referred to has been changed to 'referral for goals relating to' for clarity. (p5)